# Deep-Sea Epibenthic Megafaunal Assemblages of the Falkland Islands, Southwest Atlantic

**T. R. R. Pearman** [1,*] **, Paul E. Brewin** [1,2] **, Alastair M. M. Baylis** [1] **and Paul Brickle** [1,3]

1    South Atlantic Environmental Research Institute (SAERI), Stanley FIQQ 1ZZ, Falkland Islands
2    Shallow Marine Surveys Group, Stanley FIQQ 1ZZ, Falkland Islands
3    School of Biological Sciences (Zoology), University of Aberdeen, Tillydrone Avenue, Aberdeen AB24 2TZ, UK
*    Correspondence: tpearman@saeri.ac.fk

**Abstract:** Deep-sea environments face increasing pressure from anthropogenic exploitation and climate change, but remain poorly studied. Hence, there is an urgent need to compile quantitative baseline data on faunal assemblages, and improve our understanding of the processes that drive faunal assemblage composition in deep-sea environments. The Southwest Atlantic deep sea is an undersampled region that hosts unique and globally important faunal assemblages. To date, our knowledge of these assemblages has been predominantly based on ex situ analysis of scientific trawl and fisheries bycatch specimens, limiting our ability to characterise faunal assemblages. Incidental sampling and fisheries bycatch data indicate that the Falkland Islands deep sea hosts a diversity of fauna, including vulnerable marine ecosystem (VME) indicator taxa. To increase our knowledge of Southwest Atlantic deep-sea epibenthic megafauna assemblages, benthic imagery, comprising 696 images collected along the upper slope (1070–1880 m) of the Falkland Islands conservation zones (FCZs) in 2014, was annotated, with epibenthic megafauna and substrata recorded. A suite of terrain derivatives were also calculated from GEBCO bathymetry and oceanographic variables extracted from global models. The environmental conditions coincident with annotated image locations were calculated, and multivariate analysis was undertaken using 288 'sample' images to characterize faunal assemblages and discern their environmental drivers. Three main faunal assemblages representing two different sea pen and cup coral assemblages, and an assemblage characterised by sponges and Stylasteridae, were identified. Subvariants driven by varying dominance of sponges, Stylasteridae, and the stony coral, *Bathelia candida*, were also observed. The fauna observed are consistent with that recorded for the wider southern Patagonian Slope. Several faunal assemblages had attributes of VMEs. Faunal assemblages appear to be influenced by the interaction between topography and the Falkland Current, which, in turn, likely influences substrata and food availability. Our quantitative analyses provide a baseline for the southern Patagonian shelf/slope environment of the FCZs, against which to compare other assemblages and assess environmental drivers and anthropogenic impacts.

**Keywords:** vulnerable marine ecosystems; multivariate analysis; cold-water corals; sponges

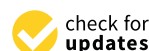



## 1. Introduction

Deep-sea environments (water depth > 200 m) are under increasing threat from anthropogenic pressure [1–5], including climate change [6–9]. As a result, improving the management and conservation of deep-sea environments is frequently identified as a priority action [1,6,10–14]. However, deep-sea environments are poorly understood [10,15,16], especially in the Southwest Atlantic, despite it being increasingly advocated as a biodiverse region [17–21] that supports endemic species [22,23] and globally important ecosystems [24].

To date, deep-sea faunal exploration of the Southwest Atlantic has predominately centred on the Brazilian shelf and slope [21,25–28]. In contrast, the faunal composition of the southern Atlantic Patagonian deep-sea is largely unknown [18–20,29], and our current knowledge of the epibenthos is predominately based on ex situ data—fisheries bycatch or

scientific trawl/dredge data [17,30–42], with most records from Discovery Expeditions of the early twentieth century [43]. Only limited inference can be drawn from ex situ data, because sample bias prevents quantitative characterisation of faunal assemblages [44]. As a result, our knowledge of the epibenthos is restricted to species inventories, single-species descriptions [18,19,38,39,45–48], and qualitative/semiquantitative assemblage descriptions at the level of phyla [17,32,33,37,49]. Underwater imagery offers a non-invasive method to undertake in situ quantitative characterisation of epibenthic assemblages [44,50–59]. However, the few studies utilising benthic imagery of Southwest Atlantic deep-sea epibenthos have remained descriptive [29,42,49,60–68], and the one study that undertook quantitative analysis did so at the level of phyla, preventing detailed characterisation of assemblages [69]. The lack of quantitative studies leaves a gap in our understanding of whether observed assemblages are statistically robust entities, preventing comparisons with other deep-sea localities [70] and limiting our ability to identify environmental drivers of assemblage composition. This knowledge is necessary if we are to implement efficient management practices to safeguard deep-sea assemblages from anthropogenic impacts, including climate change [71,72].

Describing deep-sea epibenthic megafaunal assemblages is especially important because they predominantly comprise long-lived [73–75], fragile species that exhibit low resilience to environmental impacts [76,77] and slow recovery rates [77,78]. These attributes contribute to species and assemblage vulnerability, and therefore, many deep-sea species, assemblages and habitats are considered components of vulnerable marine ecosystems (VMEs) [12]. Vulnerable marine ecosystems are internationally recognised as ecologically important, and legislation to identify and map VMEs to avoid significant adverse impacts has been developed [11,12].

The Falkland Islands are an archipelago located at the southeast extremity of the Patagonian shelf in the Southwest Atlantic. The deep sea constitutes approximately 72% of the marine environment encompassed by the Falkland Islands conservation zones (FCZs; comprising of the Falklands Interim Conservation and Management Zone and the Falklands Outer Conservation Zone). However, little is known of the deep-sea faunal assemblages occurring within the FCZs. The few published studies report species records collected as part of wider Southwest Atlantic surveys and/or macroinfaunal analysis reported in the grey literature [60–67]. Published species records indicate that the deep-sea FCZs hosts diverse fauna, including VME indicator taxa [22,36]. However, to date, no quantitative analysis of epibenthic megafauna has been published. In light of the proposed Falkland Islands marine managed areas (MMAs) public consultation in 2022, there is a pressing need to improve our baseline knowledge of deep-sea assemblages, including VMEs occurring in the FCZs, so that appropriate management practices to safeguard VMEs can be developed.

To increase our bioecological knowledge of Southwest Atlantic deep-sea epibenthic megafaunal assemblages, including VMEs, we combined legacy image datasets previously collected for commercial purposes, with freely available environmental data constituting bathymetry and oceanography datasets to (1) characterise the composition of epibenthic megafauna assemblages and (2) investigate the environmental factors that might influence epibenthic megafauna community structure.

## 2. Materials and Methods

### 2.1. Study Area

The Falkland Islands are situated on an area known as the Falkland Plateau, an extension of the Patagonian shelf, reaching water depths of 2500 m. The Falkland Plateau is separated to the north from the Argentine Basin by the Falkland Escarpment, and to the south, the Falkland Trough separates the Falkland Plateau from the Burdwood Bank (Figure 1).

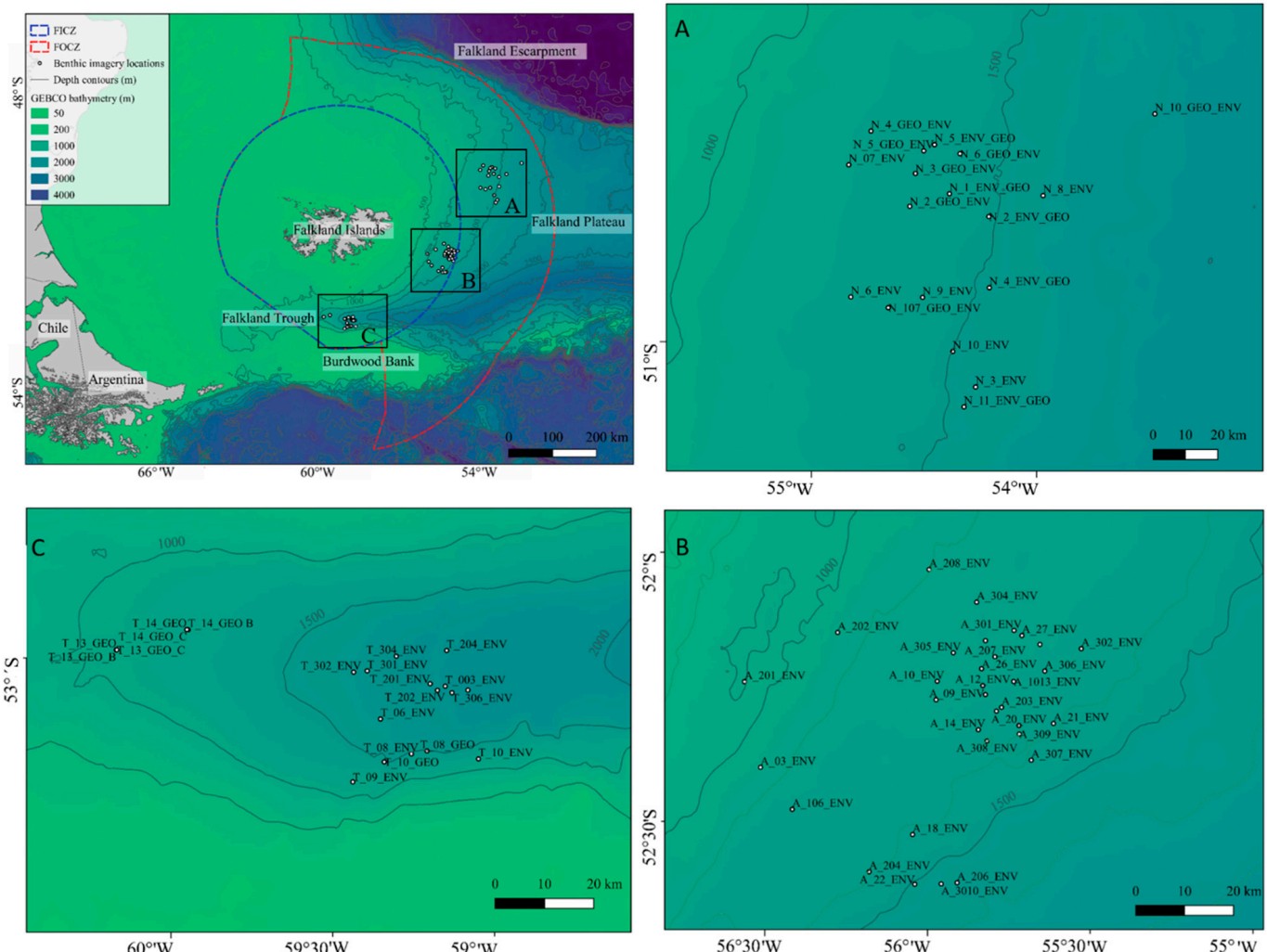

**Figure 1.** Location map of the Falkland Islands and (**A–C**) zoomed insets of drop-down camera locations (indicated as black circles). The Falklands conservation zones encompass the Falklands Interim Conservation and Management Zone (FICZ) and the Falklands Outer Conservation Zone (FOCZ). (See Supplementary Table S1 for full station list and locations).

The Falkland Islands are considered part of the cold–temperate Atlantic Magellan Biogeographic Province [79–81], and the more recently assigned cold–temperate South American Biogeographic Region [82], within which it is proposed the islands form a sub-province with Southern Argentina [83]. Circulation in the region is characterised by a northeastward flow that extends from the tip of Tierra del Fuego to the subtropical shelf front near Uruguay, from where it veers southeast into the deep sea [84]. The Falkland Current extends along the Patagonian shelf break and upper slope until the subtropical shelf front. In the FCZs, the Falkland Current diverges around the Falkland Islands, forming an eastern and western stream [84]. The Falkland Current provides a constant influx of sub-Antarctic waters [84], and has a mean temperature range of 4–11 °C, salinity range of 33.8–33.4 psu, and is associated with high primary productivity, which, in turn, supports important fisheries [85]. The slope waters of the FCZs are characterised by the Antarctic Intermediate Water (AAIW) at water depths < 1000 m, and the Upper Circumpolar Deep Water (UCDW) between water depths of 1000–2200 m.

The substrata along the Patagonian shelf mainly consist of mud, muddy sands, sandy muds, and pebbly muds [42,86,87], with mid-water drifts on the slope composed of hemipelagic mud and ice-rafted debris [87]. The Falkland Islands are considered regionally unique in that they did not experience glaciation during the last glacial period, resulting

in a benthic environment that has been permanently sustained. Still, the influence of past geological periods is evident in the presence of iceberg plough marks and ice-rafted debris [88].

### 2.2. Quantitative Image Analysis

Benthic imagery was obtained from 69 stations (Figure 1), collected during three environmental baseline surveys conducted by Noble in 2014 (see Supplementary Table S1; Falkland Islands Government Department of Mineral Resources, unpublished data) that covered water depths of 1070 m to 1880 m. Imagery data were collected using a "Sea Bug" drop-down camera system equipped with a standard definition stills camera (G10 Canon, five megapixels) and a laser scale with parallel beams positioned 30 cm apart. Positional data were derived from an ultra-short baseline navigation system (USBL) beacon attached to the cable above the camera frame. At each station, the drop-down camera system was towed for 100 m at approximately 0.5 knots.

A total of 862 georeferenced benthic still images were reviewed, and images of low visibility (due to sediment or lighting) or where laser points were not visible, were removed. To ensure overlapping images were not analysed, and to reduce the influence of spatial autocorrelation, "Sample" images were annotated at 60 s intervals. All epibenthic megafauna >10 mm were enumerated and identified to the lowest taxonomic level possible, and assigned to morphospecies (visually distinct taxa) in BIIGLE 2.0 platform [89]. The assignment of morphospecies when analysing image data is common practice in deep-sea settings where access to specimens and a good knowledge of taxonomy is absent [90,91]. The area of each image was calculated and used to convert counts to densities per m$^2$. The area of each image was calculated using the pixel dimension of each image together with the pixel and actual distance between lasers [92]. The mean image area was 1.14 m$^2$ $\pm$ 0.95.

To explore assemblage–environment relationships, a dominant substrata type (based upon EUNIS 2022 classifications [93] (see Supplementary Figure S1) was assigned to each annotated image. To capture the influence of geomorphology and oceanography acting at broad spatial scales, a suite of terrain derivatives were calculated from GEBCO bathymetry gridded at ~430 m (0.004°) using ArcGIS extension Benthic Terrain Modeler v. 3.0 (Table 1). Oceanography variables were extracted from global models and exported as rasters interpolated to a resolution of ~430 m (0.004°) by kriging using the Spatial Analyst toolbox in ArcGIS (Table 1).

**Table 1.** Environmental variables used in modelling. † Environmental variable retained in final model.

| Environmental Variables | Description | Unit | Native Resolution |
|---|---|---|---|
| Seabed terrain | | | |
| Depth † | Bathymetry extracted from GEBCO. https://www.gebco.net/ (accessed on 10 November 2020) | m | 0.004° |
| Terrain derivatives | | | |
| Slope † | A first derivative of bathymetry measuring the change in elevation from one pixel to its neighbour derived from a neighbourhood size of 3 × 3 | ° | 0.004° |
| Eastness † | A first derivative of bathymetry measuring the easterly orientation of maximum change along the slope on a continuous scale (−1 to +1) | - | 0.004° |
| Northness † | A first derivative of bathymetry measuring the northerly orientation of maximum change along the slope on a continuous scale (−1 to +1) | - | 0.004° |
| Curvature | A second derivative of bathymetry measuring the shape of the slope, with values indicating whether a slope is convex or concave | - | 0.004° |

**Table 1.** *Cont.*

| Environmental Variables | Description | Unit | Native Resolution |
|---|---|---|---|
| Fine bathymetric position index (FBPI) † | A derived metric of a cell's position and elevation relative to its surrounding landscape/cells within a user defined area [94] | - | |
| Broad bathymetric position index (BBPI) | A derived metric of a cell's position and elevation relative to its surrounding landscape/cells within a user defined area [94] | - | |
| Rugosity | A measure of the ratio of the surface area to the planar area calculated with a neighbourhood size of $3 \times 3$ pixels [95] | - | |
| Oceanography variables | | | |
| Surface chlorophyll | Extracted from GLOBAL_REANALYSIS_BIO_001_029 http://marine.copernicus.eu (accessed on 10 November 2020) | mg m$^{-3}$ | 0.08° |
| Seabed temperature | Extracted from GLOBAL_ANALYSIS_FORECAST_PHY_001_024 http://marine.copernicus.eu (accessed on 10 November 2020) | °C | 0.08° |
| Mean seabed current velocity U | Extracted from GLOBAL_ANALYSIS_FORECAST_PHY_001_024 http://marine.copernicus.eu (accessed on 10 November 2020) | m/s | 0.08° |
| Seabed Ph | Extracted from GLOBAL_REANALYSIS_BIO_001_029 http://marine.copernicus.eu (accessed on 10 November 2020) | - | 0.08° |
| Surface Ph | Extracted from GLOBAL_REANALYSIS_BIO_001_029 http://marine.copernicus.eu (accessed on 10 November 2020) | - | 0.08° |
| Seabed phosphate | Extracted from GLOBAL_REANALYSIS_BIO_001_029 http://marine.copernicus.eu (accessed on 10 November 2020) | μmol kg$^{-1}$ | 0.08° |
| Surface phosphate | Extracted from GLOBAL_REANALYSIS_BIO_001_029 http://marine.copernicus.eu (accessed on 10 November 2020) | μmol kg$^{-1}$ | 0.08° |
| Surface dissolved oxygen | Extracted from GLOBAL_REANALYSIS_BIO_001_029 http://marine.copernicus.eu (accessed on 10 November 2020) | μmol kg$^{-1}$ | 0.08° |
| Seabed silicate | Extracted from GLOBAL_REANALYSIS_BIO_001_029 http://marine.copernicus.eu (accessed on 10 November 2020) | μmol kg$^{-1}$ | 0.08° |
| Aragonite saturation state | Extracted from GLODAPv.2.2016b [96] (accessed on 22 March 2018) | μmol kg$^{-1}$ | 1° |
| Dissolved inorganic carbon | Extracted from GLODAPv.2.2016b [96] (accessed on 22 March 2018) | μmol kg$^{-1}$ | 1° |
| Calcite saturation state | Extracted from GLODAPv.2.2016b [96] (accessed on 22 March 2018) | μmol kg$^{-1}$ | 1° |
| Nitrate | Extracted from GLODAPv.2.2016b [96] (accessed on 22 March 2018) | μmol kg$^{-1}$ | 1° |
| Total alkalinity | Extracted from GLODAPv.2.2016b [96] (accessed on 22 March 2018) | μmol kg$^{-1}$ | 1° |
| Substrata variables | | | |
| Substrata † | Substrate type annotated from imagery based upon EUNIS 2022 classifications [93] | | |

*2.3. Data Analysis*

Multivariate analysis was used to identify faunal assemblages and environmental variables influencing assemblage composition. Prior to multivariate analysis, sample images with no or fewer than three taxa were removed to reduce the influence of rare taxa that may reflect sampling artifacts associated with deep-sea sampling rather than representing the whole community [97]. Of the 694 images annotated, 195 had no visible fauna and 213 images had less than three taxa, resulting in 288 sample images retained for multivariate analysis. Environmental data coincident with each sample image location were extracted from the environmental rasters and combined with the substratum annotation of that sample.

Epibenthic megafaunal assemblages were assessed by non-metric multidimensional scaling (nMDS) and hierarchal cluster analysis with group-averaged linkage, using a Hellinger dissimilarity matrix derived from the Hellinger-transformed data matrix. A Hellinger transformation was used to enable the use of linear ordination methods in the canonical

redundancy analysis (RDA) [98,99]. The optimal number of interpretable clusters was determined with fusion level and mean silhouette widths [99]. Characteristic morphospecies contributing to similarity among clusters were identified using similarity percentage analysis (SIMPER) [100]. To explore the relationships between multivariate morphospecies data and different environmental variables, RDA was performed [99]. RDA combines the outputs of multiple regression with ordination. To obtain the most parsimonious model, forward selection was carried out on the standardised (i.e., transformed to zero mean and unit variance) environmental variables, and Pearson's correlation and variance inflation factor (VIF) scores were used to exclude collinear environmental variables in the model (correlation coefficients >0.7) [101]. High collinearity in environmental variables during model selection resulted in a subset of variables being retained (fine bathymetric position index (FBPI), slope, depth, mean current velocity, and aspect—which was separated into eastness and northness) (Table 1). Depth *per se* does not influence fauna, but was retained as it is correlated with quantified and unquantified environmental variables (water mass properties, food availability) that have been shown to influence deep-sea faunal assemblage patterns [102–104].

All statistical analyses were conducted using the open source software R [105] packages "Packfor" "vegan", "cluster", "ape", "ade4", "gclus", "AEM", "spdep", and "MASS".

## 3. Results

In total, 7398 individuals belonging to 157 morphospecies from eight known phyla were annotated from the 694 sample images (see Supplementary Table S2). Many morphospecies were rare, being observed from a few images at low abundance, and only 17 morphospecies were observed from more than 10% of sample images (see Supplementary Table S2). The hydrocoral Stylasteridae sp. was the most abundant and commonly encountered morphospecies across samples, occurring across 56.2% of samples and representing 15.5% of total individuals observed (see Supplementary Table S2). Porifera were also commonly observed, with the morphospecies referred to as Massive Ball Porifera occurring cross 28.1% of the sample images and representing 4.3% of total individuals, and encrusting Porifera occurring cross 20.9% of the sample images and representing 3.8% of total individuals (see Supplementary Table S2).

Soft, muddy, and sandy substrata dominated samples from the slope of the Falkland Trough from which fields of sea pens, including *Anthoptilum grandiflorum* and solitary cup corals of the genus *Flabellum*, were observed (Figure 2E). The substrata from the Falkland Plateau were more varied, encompassing muddy and sandy substrata with varying degrees of pebbles, glacial dropstones, and hard substrata. Sea pens and cup corals also occurred in these soft substrata (Figure 2D,E). On the other hand, sponges (predominantly belonging to Hexactinellida and Demospongiae) and Stylasteridae occurred across coarse substrata (Figure 2C), interspersed with non-reef aggregations of the stony coral, *Bathelia candida*, soft corals of the Alcyonacea (including formally Gorgonacea), large Stylasteridae such as *Stylaster densicalus* and erect Porifera, including *Phakellia* sp. that occurred on coarser substrata and glacial dropstones (Figure 2B). Near escarpments *B. candida* also formed reef-like aggregations with their coral framework and rubble, colonised by encrusting Porifera, Stylasteridae, the soft coral *Thouarella viridis*, and the squat lobster *Munida spinosa* (Figure 2A).

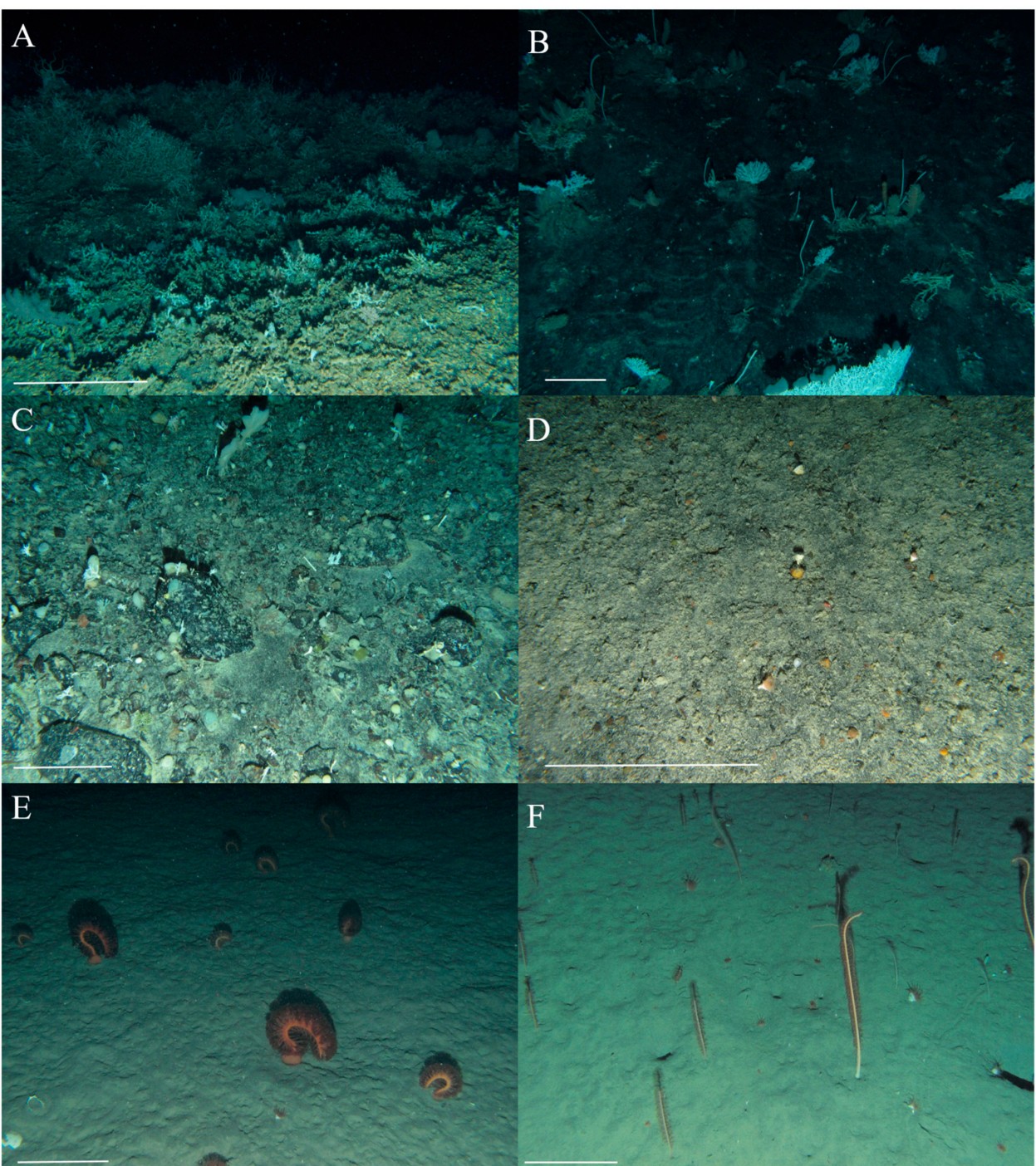

**Figure 2.** Example images of fauna and substrata. (**A**) *Bathelia candida* reef observed at 1280 m water depth. (**B**) Mixed cold-water corals, including *Stylaster densicaulis* and Primnoidae, Stylasteridae, and Porifera morphospecies observed at 1280 m water depth. (**C**) Stylasteridae and Massive Ball Porifera morphospecies were observed from mixed substratum at 1595 m water depth. (**D**) Solitary cup corals, including *Flabellum* sp. and Scleractinia sp. 5, were observed from coarse substratum at 1406 m water depth. (**E**) *Anthoptilum grandiflorum* and *Flabellum* sp. observed from sandy mud at 1225 m water depth. (**F**) Pennatulacea sp. and *Flabellum* sp. observed from sandy mud at 1330 m water depth. Scale bar = 30 cm.

### 3.1. Quantitative Analysis of Epibenthic Megafaunal Assemblages

Fusion level and mean silhouette widths identified five clusters as optimal in representing faunal assemblages from the 288 sample images used in hierarchal clustering (see Supplementary Table S2 and Figures S2 and S3). The superimposed groupings from the hierarchal analysis onto the nMDS plot show a general agreement between the two methods (Figure 3), with assemblages distributed across the FCZ (Figure 4). On the other hand, the RDA analysis highlights further differentiation within cluster one (Figure 5). Cluster one represents the most commonly encountered assemblage type (Figures 3 and 4) characterised by a predominance of Stylasteridae and sponges (Figures 2A–C and 3 and Table 2). Cluster two and three represent variants of the sea pen and solitary cup coral assemblages. Cluster two is characterised by *Anthoptilum grandiflorum* and an attached solitary cup coral, Scleractinia sp. 5 (Figures 2D,E and 3 and Table 2). In contrast, cluster three is characterised by, Pennatulacea sp. and a predominance of the unattached solitary cup coral, *Flabellum* sp. (Figures 2F and 3 and Table 2). Alcyonacea sp. 19 and Hormathiidae sp. 2 characterised cluster four, and cluster five was characterised by the morphospecies cold-water whip coral (Figure 3 and Table 2). However, both these clusters were only represented by two sample images, limiting conclusions that can be drawn, and so are omitted from further discussion.

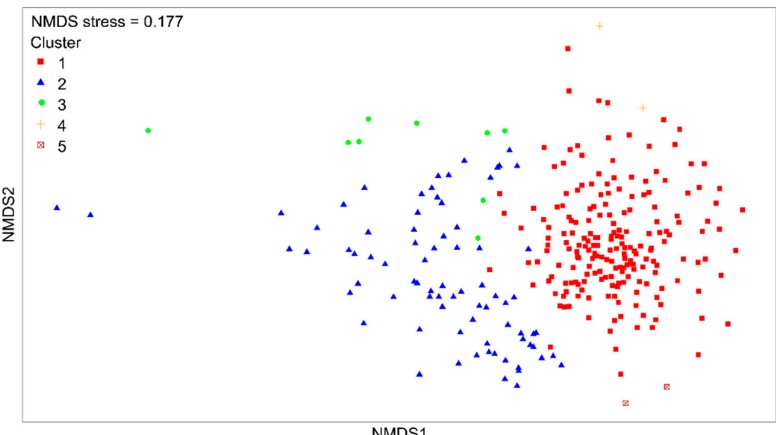

**Figure 3.** nMDS plot of multivariate Hellinger distance matrix of transformed morphospecies density data. Samples are coloured to represent the five clusters identified by hierarchal clustering analysis.

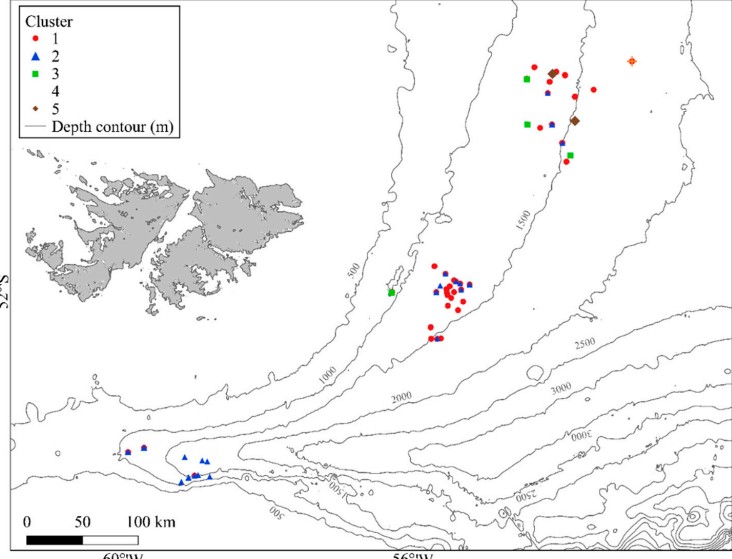

**Figure 4.** Spatial distribution of clusters identified from hierarchal clustering of Hellinger distance matrix of transformed morphospecies density data.

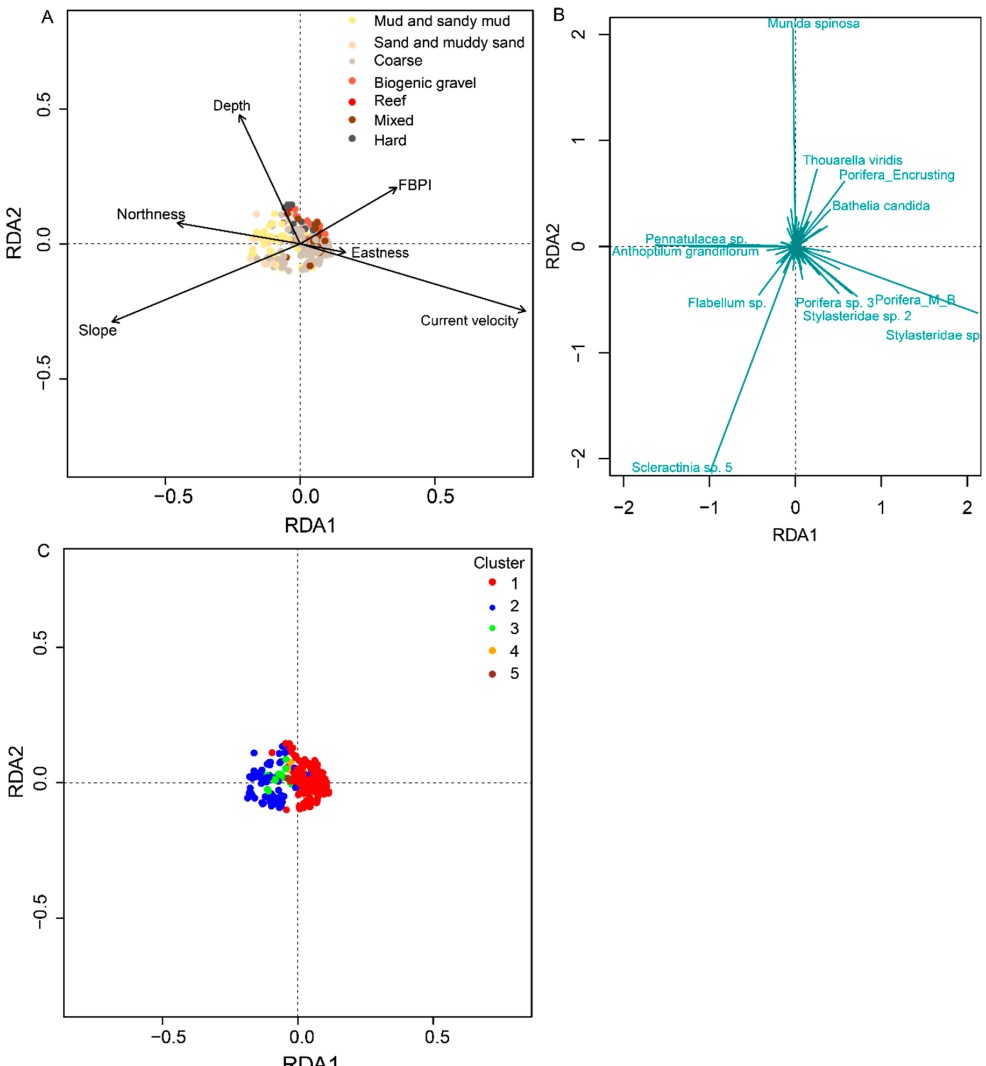

**Figure 5.** Canonical redundancy analysis of a Hellinger distance matrix of transformed morphospecies density data and selected environmental variables. For clarity, the triplot is displayed in three separate plots with varying axis limits. (**A**) Environmental variables and sites colour coded by substratum type. (**B**) Morphospecies data, only fauna with the strongest effect are labelled. (**C**) Sites colour coded to represent hierarchal clustering. The vector arrowheads represent high, the origin averages, and the tail (when extended through the origin) low values of the selected environmental variables. Sites are represented by circles. Sites close to one another tend to have similar faunal composition that those further apart.

**Table 2.** Clusters identified from multivariate hierarchal clustering analysis with associated environmental parameters, number of samples represented by each cluster (N), and SIMPER results identifying the morphospecies that characterise the clusters (70% accumulative contribution cut-off).

| Cluster | Characterising Morphospecies | Water Depth (m) | Substratum | N |
|---------|------------------------------|-----------------|------------|---|
| 1 | Stylasteridae sp., Massive Ball Porifera | 1070–1840 | Coarse, biogenic gravel, reef, mixed | 197 |
| 2 | Scleractinia sp. 5, *Anthoptilum grandiflorum* | 1070–1880 | Coarse, mud, sandy mud Sand, muddy sand | 78 |
| 3 | *Flabellum* sp., Pennatulacea sp. | 1120–1327 | Mud, sandy mud | 9 |
| 4 | Alcyonacea sp. 19, Hormathiidae sp. 2 | 1840 | Hard | 2 |
| 5 | Cold-water whip coral | 1390–1540 | Mud, sandy mud, coarse | 2 |

The RDA analysis shows that the vectors representing species scores (Figure 5) separate into three main subgroups. The upper right quadrant is characterised by the predominance of *M. spinosa*, encrusting Porifera, *T. viridis*, and *B. candida*. The lower right quadrant is represented by a predominance of Stylasteridae and Porifera morphospecies, and the lower left quadrant is represented by the predominance of *A. grandiflorum*, Pennatulacea sp., Scleractinia sp. 5, and *Flabellum* sp. morphospecies. Within the lower left quadrant, there was further differentiation between sea pen and cup coral predominance.

The RDA analysis, hierarchal clustering, and nMDS plot showed similar trends in that the main clusters separated into two regions, comprised of the same characterising morphospecies (Figures 3 and 5). Cluster one relates to the right quadrants, and cluster two and three relate to the lower left quadrant, within which differentiation between cluster two and three, driven by relative abundance of Scleractinia sp. 5, could be seen (Figure 5). However, the RDA analysis indicated a further level of differentiation, within cluster one of the hierarchal clustering, separating the encrusting Porifera, *B. candida*, *T. viridis*, and *M. spinosa* assemblage from that dominated by Stylasteridae and Massive Ball Porifera (Figure 5). The inability of the hierarchal clustering to differentiate between these latter two assemblages likely arises from the commonality of Stylasteridae, which led to cluster cohesion (Table 2) and the observation that assemblages occur as continuums rather than distinct entities distributed next to one another, which reduces the discriminatory power of hierarchal clustering [101].

### 3.2. Environmental Drivers of Faunal Assemblages

The RDA analysis shows that current velocity, topography (depth, aspect, slope, and FBPI), and substrata influence faunal assemblages (adjusted $R^2$ 13%) (Figure 5 and Table 3). The first axis of the RDA plot (Figure 5) represents a gradient from sandy muds to coarse, and then hard (including reef) substrata, with increasing current velocity, and from relatively smooth broad-scale sloping topography to more rugged flatter topography. The second axis of the RDA plot represents a depth gradient.

**Table 3.** Results from canonical redundancy analysis (RDA) of Hellinger-transformed species data and selected environmental variables. Significance of individual terms determined by analysis of variance (ANOVA) on RDA. *** $p \leq 0.001$.

| Environmental Variables—Significance of Individual Terms by ANOVA | Adjusted $R^2$ | Significance of RDA Plot by ANOVA | |
|---|---|---|---|
| | | F-Value | *p*-Value |
| Depth ***, Slope ***, FBPI ***, Substrate ***, Eastness ***, Northness ***, Current Velocity *** | 13 | 4.51, df = 12,275 | 0.001 |

## 4. Discussion

### 4.1. Deep-Sea Epibenthic Megafaunal Assemblages of the Falkland Islands

Our multivariate analysis has enabled the first quantitative characterisation of deep-sea epibenthic megafaunal assemblages within the FCZs, and contributes toward our understanding of environmental factors influencing southern Patagonian deep-sea assemblages.

Quantitative analysis identified three main assemblages that are composed of taxa with southern Patagonian shelf/slope distributions, and are characterised by fragile habitat-forming taxa considered indicators and/or components of VMEs [12,106].

Although *B. candida* was not identified as a characteristic species of cluster one, it was observed forming reef-like structures with an associated fauna of *M. spinosa*, *T. viridis*, and encrusting Porifera (Figure 2A), depicted as species vectors diverging from the rest of cluster one in the RDA plot (Figure 5). *Bathelia candida* is a framework-forming Scleractinian with a Southwest Atlantic distribution, occurring in the offshore waters of southern South America from Rio Grande to Southern Chile [22,23]. *Bathelia candida* is reported along the Patagonian shelf and slope [22–24,29,37], with recent discoveries of reefs described

from the nearby Perito Moreno Terrase [29,31,40,42,49] and a coral mound province from Argentina (Northern Argentine Mound Province) [24]. In the FCZs, *B. candida* was observed as single discrete colonies and forming continuous framework reef structures (Figure 2A). *Bathelia candida* is a less-known framework-forming Scleractinian compared to *Lophelia pertusa* (recently synonymised to *Desmophyllum pertusum* [107]), *Madrepora oculata*, and *Solenosmilia variabilis* [97,108,109]; our observations add to the known distribution of this species and to our knowledge, also represent the most southerly record of reef habitat for *B. candida* in the Southwest Atlantic.

*Bathelia candida* was observed forming reef-like structures that are of ecological importance in the area. Cold-water coral reefs are generally regarded as long-lived [73], relatively slow growing [73,77,78], structurally complex [23,31,110,111], and functional significant habitats that, due to their life history characteristics, exhibit high fragility to fishing impacts [76], and as such, are recognised as VME habitats. Cold-water coral reefs increase structural complexity provided by living and dead coral frameworks, which provide a hard substrate that increases environmental heterogeneity, resulting in increased biodiversity of associated fauna [109,110]. Cold-water coral reefs are also nursery habitats for numerous species, including commercially important fish species [97,112–116]. In our study, corals, sponges and bryozoa were observed growing on *B. candida* frameworks and rubble (Figure 2A), while *M. spinosa* were seen beneath rubble. These observations are consistent with descriptions and bycatch records from the Patagonian shelf edge and slope [31,32,37,40,42,49]. Similarly, corals, sponges, and bryozoa have been observed on *D. pertusum* reefs [52,117] in the Northeast Atlantic, and *Munida* squat lobsters have been observed seeking refuge beneath *D. pertusum* coral rubble [54,118]. Little is known of the ecology of *B. candida* reefs, but our observations of similar faunal associations support the hypothesis that their functional role is analogous to that of *D. pertusum* reefs [29], and therefore, *B. candida* reefs represent important features within the FCZs.

*Bathelia candida* was also observed forming non-reef aggregations with Alcyonacea (including formally Gorgonacea), large Stylasteridae, and erect Porifera. These "hard-bottomed coral garden" VMEs were commonly observed on coarser substrata and glacial dropstones (Figure 2B). Coral gardens comprising similar taxa to those observed in our study have been described from the Patagonian shelf edge and slope [17,29,31,37,38,40,42,49], including the Burdwood Bank [17,37,38], indicating a continuous southern Patagonian distribution of this assemblage, for which the Burdwood Bank and FCZs likely act as source locations connected to northern populations via the northeastward flow of the Falkland Current.

There were certain areas where some characteristic species of cluster one exhibited higher dominance and abundances (Table 1 and Figure 5). Stylasteridae and sponges (predominantly belonging to Hexactinellida and Demospongiae) were observed across coarse substrata, and, in certain areas, exhibited higher dominance and abundances (Figure 2C and Table 1), depicted as species vectors diverging from the rest of cluster one in the RDA plot (Figure 5). Sponge aggregations have been recorded from the Patagonian shelf [31,37,40,42,49], and are generally a common component of epifaunal assemblages that often represent the dominant taxa by biomass [37,119,120]. "Deep-sea sponge aggregations" are considered VMEs [77,121] due to their functional role in benthic–pelagic coupling via carbon and nutrient cycling [122,123] and their structural complexity, which can enhance and create structurally complex habitats [91,124–128]. In turn, these habitats increase associated biodiversity, and provide nurseries and refuge [51,91,129]. On the other hand, sponge fragility, longevity [77,130,131], and other life history characteristics hinder their ability to recover [77], making them vulnerable to anthropogenic impacts. Sponge morphology has been shown to influence the composition, diversity, and abundance of associated fauna [91,126,132]. In our study, the most frequently annotated sponges were Massive Ball morphotypes, which have relatively low structural complexity. However, it is possible that at specific densities, these sponges may act together with sparser erect branching sponges to increase habitat structural complexity [51,91,128,132,133]. Sponge densities exceeding

the OSPAR threshold for designating sponge aggregations (0.5–24 sponges/m$^2$) [133] were recorded from the study area (Supplementary Table S2), and often coincided with increased sponge diversity. However, OSPAR criteria require that sponges represent the dominant characterising taxa within the assemblage [134], whereas in our study, Stylasteridae, which co-occurs with sponges, is also a characteristic taxa of the assemblage (Figure 5 and Table 2). Further research will be required to ascertain the functional significance of this assemblage in the FCZ, and whether it meets the criteria of a "Deep-sea sponge aggregation" VME.

The sea pen and solitary cup coral assemblages represented by clusters two and three (Figure 5 and Table 2) were observed from soft substrata (Figure 2D–F). Solitary cup corals predominantly occupy upper slope environments, and *Flabellum*-dominated "cup coral fields" have been recorded along the Patagonian slope [23,32,49], while other cup coral species exhibit southern Patagonian restricted ranges [23]. Cup coral fields are considered a type of "soft-bottomed coral garden" VME habitat [31,42,77]. Little is known of the ecology of Southwest Atlantic cup coral fields. However, it is likely that Southwest Atlantic "cup coral fields" support similar functional roles as "solitary Scleractinian fields" described from slope environments of the Northeast Atlantic [135].

In our study, the sea pen Pennatulacea sp. could not be discerned to species level from imagery, but resembled the genus *Balticina* (formally *Halipteris*). *Anthoptilum grandiflorum* and *Balticina* have circumglobal deep-sea distributions, and have been recorded along the Patagonian shelf and slope, including the northwest slope of the Burdwood Bank [17,39,42]. "Sea pen fields" have previously been described from the FCZs [36], and inferred from trawl bycatch off of the Argentine slope [32] and northwest slope of the Burdwood Bank [17,39]. Sea pen fields are considered VME habitats [31,42,77] because they are long-lived, slow growing, fragile taxa, and can form meadows that [136–138] act as nurseries [139] and increase local biodiversity [136] by providing structural complexity and habitat heterogeneity [110]. In the Northwest Atlantic, 14 species of sea pen were associated with *A. grandiflorum* and *H. finmarchica* [39,136], while in the Southwest Atlantic, the anemone *Hormathia pectinata* is found growing on *A. grandiflorum* [39]. Brewin et al., (2020) conducted a spatial analysis of predicted sea pen distributions in relation to fishing footprints, and suggested that sea pen assemblages have a restricted distribution within the FCZ, and as such, are most vulnerable to fishing. Their study modelled sea pen fields as a monospecific habitat, whereas our study has shown that there is variation between sea pen assemblages based on the dominant sea pen and cup coral species present (Figures 2–5 and Table 2). The fact that sea pen fields of the FCZs are not comprised of a monospecific assemblage further increases their vulnerability, as the spatial extent of each variant is less than predicted when modelled as a single assemblage; therefore, our findings should be considered in future fishing impact assessments.

In addition to the main assemblages identified from our analysis, VME indicator taxa indicative of VMEs, namely "bryozoan patches", "tube-dwelling anemone patches comprised of Cerianthidae", and "chemosynthetic communities", were also observed (see Supplementary Table S2). However, these assemblages were not differentiated from the broader assemblages (Figures 3 and 5), probably because they did not form large aggregations or patches, and due to the commonality of Stylasteridae and Porifera morphospecies that maintained cluster similarity (see Supplementary Table S2 and Figure S3).

Our analysis of epibenthic megafauna is based upon the annotation of morphospecies. Morphospecies enable taxa to be differentiated beyond the level of taxonomic hierarchy achievable using traditional taxonomic features that cannot be discerned from imagery. Distinguishing morphologically distinct taxa (which are therefore likely to be taxonomically distinct) preserves important information on biodiversity [90]. However, there are limitations with using morphospecies as units because faunal groups with similar gross morphology, which rely upon microscopic features to distinguish species, can result in multiple species being annotated as a single morphospecies. In our study, the morphospecies that are most likely to have been susceptible to this bias are encrusting Porifera and Massive Ball Porifera because many sponge species have adopted this gross morphology. Despite

the potential of "clumping" species together within a single morphospecies, our analysis discerned differences in megafauna assemblages that are characterised by taxa that differ at high taxonomic levels, where the chance of misidentification is lower (i.e., Pennatulacea versus Stylasteridae), and thus, still provides useful insight into megafaunal assemblages of the FCZ. To support further research, future efforts should look toward collaborative imagery databases that will enable consistency in annotation of morphospecies [90], and species identifications from imagery should be verified with specimens when possible.

### 4.2. Environmental Drivers of Faunal Assemblages

Knowledge of broad-scale environmental drivers of deep-sea faunal assemblages is relatively well established [103,140–142]. However, our understanding of how these environmental drivers interact and influence deep-sea faunal assemblages in the Southwest Atlantic is poorly understood. Our lack of knowledge is, in part, reflective of the scant available data that limits research. For example, in our study, the low variance explained in the RDA analysis (Table 3) likely reflects the broad resolution of environmental variables incorporated into the model, which inadequately discerned the fine-scale environmental variability influencing faunal assemblages [57] (i.e., patchy distribution of glacial drop-stones and fine-scale geomorphological features). Despite this, our analysis still provides useful insight into the broad-scale drivers of deep-sea faunal assemblages on the FCZs slope, and suggests that topography, substrata, and current velocity interact to influence deep-sea faunal assemblages of the FCZ slope (Figure 5 and Table 3).

Our analysis also shows a clear distinction between soft- and hard-bottom epibenthic assemblages (Figure 5), which correlated with topography (captured by the terrain variables and depth). Topography and the distribution of substrata are intrinsically linked, and influence faunal distributions [57,102,143] by providing a variety of substratum for colonisation [144,145]. Soft sediments dominate the slope of the Falkland Trough, and the relationship between this large geomorphological feature (captured by increased slope) and the soft-bottom sea pen and cup coral assemblages is shown in the RDA plot as a positive relationship between slope and mud/sand substrata (Figure 5). The topography and substrata of the upper slope of the Falkland Plateau is more complex, with evidence of contour aligned escarpments, drifts, and ice-rafted debris [62–67]. The presence of contour-aligned geomorphology and substrata indicates the influence of contour currents [146]. The Falkland Current is associated with depositional contourites and erosive features that follow the bathymetric contours on the Argentine slope [147]. The Falkland Current also flows along the slope of the Falkland Plateau [29,84], and it is therefore likely that this area experiences a similar bottom-current-controlled environment. Coral mound distributions, including those within the Northern Argentine Mound Province [24] have been directly linked to the geomorphology and substrata of regional contourite depositional systems [24,148]. In our study, the coral reef assemblage also appears to be influenced by the geomorphology and substrata, as reefs were observed from biogenic, coarse, and hard substrate associated with erosive escarpment features, which was captured in the RDA as a positive relationship between FBPI and coarser substrata (Figure 5). Filter feeders, including cold-water corals, are preferential to complex terrain [104,149–154], colonising topographic highs to exploit local current regimes, and so increase food encounter rates [154,155].

Glacial dropstones also appear to be important structures that can support "hard-bottomed coral garden" assemblages (Figure 2B). In the Northeast Atlantic, glacial dropstones occurring within soft substrata are considered "keystone structures" [57], and in a study of East Antarctic shelf fauna, dropstones were attributed to increasing diversity by increasing habitat heterogeneity [156]. In the FCZ, dropstones are distributed independent of geomorphology, and often provide hard substrata in otherwise soft substratum environments, enabling epibenthos to exploit these environments and in doing so could also be considered keystone structures.

The characteristic taxa of the assemblages observed (Porifera, Cnidaria, Hydrozoa, Alcyonacea, Scleractinia, Pennatulacea, etc.) are predominately suspension/filter feeders.

Filter feeders consume plankton and particulate organic matter (POM) [157,158], while some groups, such as sponges and bivalves, are also able to uptake dissolved organic matter [122,123] and bacteria [159]. Surface-derived POM [113,160] is an important deep-sea food resource, which in the FCZs is enhanced by the nutrient-rich Falkland Current that provides a constant influx of nutrient-rich sub-Antarctic waters along the shelf break and upper slope [29,84]. On the Argentine margin, the Falkland Current is hypothesised to facilitate cold-water coral reef growth via upwelling at the shelf, and associated surface primary production that forms an important source of POM for the fauna below [29]. In the FCZs, the interaction between the Falkland Current and underlying topography generates mesoscale eddies that facilitate the exchange of heat, momentum, and nutrients between the deep and surface waters, thus promoting surface productivity [161,162]. In our analysis, the hard-bottom coral assemblages show a positive correlation, with increased FBPI and current speed (Figure 5), and occur beneath areas of eddy formation [161], suggesting that coral assemblages of the FCZ are sustained by topography-generated eddies of the Falkland Current. This observation further supports the role of the Falkland Current in sustaining coral assemblages by enhancing food supply along the Patagonian slope [29]. Similar scenarios, where cold-water corals flourish beneath topographically modified hydrodynamics that promote and source POM, have been recorded in the Northeast [102,153,155,163–165] and Southeast Atlantic [59,165]. Undertaking further research to understand more about the Falkland Current driven surface–benthic coupling would provide invaluable information for understanding the ecology of these assemblages.

In addition to hydrodynamics, water mass properties are a key determinant of faunal composition in the deep sea [26,33,53,69,142,166,167]. Auscavitch and Waller (2017) made a comparison of faunal assemblages across the Drake Passage, including samples from the Burdwood Bank, which highlighted the influence of deep Southern Ocean water masses on assemblage structure. In the Northern Argentine Mound Province, the AAIW has been linked to coral mound development [24], and the high level of Scleractinian endism (65% of recorded species) on the Patagonian upper slope (200–1000 m) correlates with cold–temperate water masses [23]. The fauna observed during our study occur within the AAIW, and are consistent with records reported elsewhere from the southern Patagonian shelf break and slope [17,29,32,37,38,49]. This observation further supports the hypothesis of deep-sea species having a continuous distribution along the southern Patagonian shelf and slope [37], facilitated by the northeastward flow of the Falkland Current [84] and the AAIW.

## 5. Conclusions

We have undertaken the first quantitative multivariate analysis of epibenthic megafauna in the FCZs, and identified several cold-water coral and sponge assemblages, with attributes of VMEs. Our results further support the argument for a continuous southern Patagonian deep-sea fauna. We have shown that broad-scale changes in topography, substrata, and current velocity contribute to assemblage structure, and hypothesise that the Falkland Current is a key phenomenon influencing faunal assemblages on the FCZs slope. The Falkland Current promotes surface productivity and bentho–pelagic coupling, which likely sustains the observed filter and suspension feeder-dominated assemblages. Over a larger temporal scale, the Falkland Current has also acted to shape the topography and distribution of substrata, in turn, influencing faunal assemblages on the slope. To gain a greater understanding of fine-scale drivers of faunal assemblage, the collection and incorporation of higher resolution environmental data is required, and species and habitat distribution maps should be compiled. Despite the limitations of data resolution, our results still provide much needed bioecological knowledge to underpin spatial management within the FCZs, and adds to our knowledge of the biogeography of this region, which will facilitate further comparative studies of epibenthic megafauna, and support the implementation of fisheries benthic impact assessments for the Southwest Atlantic deep-sea.



**Supplementary Materials:** The following supporting information can be downloaded at: https://www.mdpi.com/article/10.3390/d14080637/s1, Figure S1: Example images of substratum type based on the EUNIS Marine Habitat Classification (https://www.eea.europa.eu/data-and-maps/data/eunis-habitat-classification-1, accessed on 10 November 2020). (A) Mixed substratum EUNIS code A6.2 observed from location N-2-GEO-ENV. (B) Hard substratum EUNIS code A6.1 observed from location N-6-GEO-ENV. (C) Coarse substratum EUNIS code A5.15 observed from location N-4-ENV-GEO. (D) Biogenic gravel (annotated as a subtype of mixed substratum) EUNIS code A6.2 observed from location N-4-ENV-GEO. (E) Muddy sand EUNIS code A6.4 observed from location N-3-ENV. (F) Mud EUNIS code A6.5 observed from location T-003-ENV. (G) Biogenic reef (adapted from EUNIS Communities of deep-sea corals that refers to reefs of *Desmophyllum pertusum*) code A6.61 observed at location A-1008-ENV; Figure S2. Hierarchal cluster analysis of Hellinger distance matrix of transformed morphospecies density data. Cluster membership is shown by the coloured rectangles. Cluster 1 = red, 2 = blue, 3 = green, 4 = orange, 5 = brown; Figure S3. Contribution (percent of total) toward total abundance of morphospecies belonging to each of the five clusters after hierarchal cluster analysis; Figure S4. Examples of morphospecies that characterise epibenthic megafaunal assemblages in the Falkland Islands Conservation Zones. Scale bar = 30 cm. Table S1: Proposed drop-down camera station in decimal degrees (°) from three environmental baseline surveys conducted by Nobel Energy in 2014. (Falkland Islands Government Department of Mineral Resources, unpublished data); Table S2: Annotated morphopecies belonging to each cluster after hierarchal cluster analysis of 288 sample images. Mean density ($m^2$) of each morphospecies within each cluster, total annotated and percent occurrence across images is provided for each morphospecies.

**Author Contributions:** Conceptualization of paper, T.R.R.P.; methodology, T.R.R.P. and P.B.; formal analysis, T.R.R.P.; data curation, T.R.R.P. and P.B.; writing—original draft preparation, T.R.R.P.; writing—review and editing, A.M.M.B., P.E.B. and P.B. All authors have read and agreed to the published version of the manuscript.

**Funding:** The research was undertaken as part of a project funded by Consolidated Fisheries Ltd., Stanley, Falkland Islands.

**Institutional Review Board Statement:** Not applicable.

**Informed Consent Statement:** Not applicable.

**Data Availability Statement:** Oceanographic data used in this study are accessible from http://marine.copernicus.eu and https://www.glodap.info/, accessed on 10 November 2020. Bathymetry is available at https://www.gebco.net/, accessed on 10 November 2020. Images are available upon request from the Falkland Islands Government Department of Mineral Resources.

**Acknowledgments:** The authors gratefully acknowledge Consolidated Fisheries Ltd. for support of this research. We thank the staff of the Falkland Islands Government Fisheries Department and Department of Mineral Resources for their support and helpful assistance in gathering various data. We thank the crews that collected the imagery data during the Noble surveys. We thank S. Cairns, N. Bax, C. Laguionie Marchais, M. Taylor, C. Goodwin and D. Morrissey for epifaunal identification help. The manuscript benefitted greatly from anonymous reviewer comments.

**Conflicts of Interest:** The authors declare no conflict of interest.

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
