# Peer review of "Deep-Sea Epibenthic Megafaunal Assemblages of the Falkland Islands, Southwest Atlantic"

_diversity, doi:10.3390/d14080637_

Round 1

Reviewer 1 Report

Dear authors,

Overall, the manuscript is interesting and well written. The data presented is valuable as it pertains to an understudied geographical area. The study design seems sound and I commend the authors on the clarity of the manuscript. However, I would like to suggest to the authors to

(1) provide further details in the methods section, specifically to clarify why only 286/862 images were used in the analysis and why a threshold of 3 taxa per image was applied here.

(2) improve the presentation of the results by adding a figure showing the density of the most prevalent species associated with each clusters and add a dbMEM analysis to assess the spatial structure of the benthic assemblages described.

I have made small comments throughout the text, which I hope will be helpful to the authors. To access these, please refer to the PDF document attached.

Author Response

July 31st

Re: Submission of revised manuscript

Dear Reviewer 1,

Thank you very much for your positive evaluation of our manuscript entitled “Deep-sea epibenthic megafaunal assemblages of the Falkland Islands, South-west Atlantic" (diversity-1792620). We would also like to thank you for your very helpful and positive feedback: we have addressed your suggestions in the revised manuscript.  A full list of our responses to each comment can be found in the attached PDF. In particular, as suggested we have expanded on the methodology for image collection and analysis in the methods. We have also included in the supplementary the density data in Table S2 and a stacked bar chart (Figure S3) to show the relative contribution of the different taxa to each assemblage. We thank you for your suggestion of including the morphospecies catalogue used in the study and we have included example images of characteristic morphospecies of each assemblage in the supplementary Figure S4. We thank you for your suggestion to consider running a dbMEM analysis to test whether epifaunal assemblages in the study region is structured at small spatial scales. This would be an interesting analysis to integrate into the study however, we do not have enough data that is spatially arranged in an appropriate manner to undertake this analysis in a meaningful way. It is however, something that we would like to explore as more of the data sets are worked up in the future.

We believe that this version of the manuscript now describes deep-sea epibenthic faunal assemblages of the Falkland Islands even better.

Sincerely,

Tabitha Pearman

Dr Tabitha R R Pearman

South Atlantic Environmental Research Institute

Stanley, Falkland Islands

Reviewer 2 Report

Dear authors,

as you mentioned in several parts of the ms, there is a scarce knowedge on the benthic communities of the Soutwest Atlantic, therefore this ms is a valuable contribution, however it needs still a bit of work. In the annotated pdf you will find my specific comments to the ms.

General comments:

please enlarge the literature research and do not ignore the research that has been done in the last decade in the South East Atlantic. The Introduction and Discussion will benefit from a wider view of the research conducted the last years.

Please revise the concept of VME as there are several mistakes in the way how you are refering to it. This is important as this kind of papers are fundamental to support the conservation of VMEs but then you need to clearly know what is a VME.

It will be nice to enlarge the literature you refer to other areas and not only the East Atlantic. There is quite a lot of work that has been done in the Mediterranean, in the North West Atlantic, South East Atlantic, Arctic, Antarctica, I know you have been refer to some but it is still highly byas to the NE Atlantic

In the mat and met section it is highly needed to add a comprehensive section about the photographic material you have been analysing, there is no information at all on the type of aterial you have been processing, area covered by the images, how did you do the annotation etc etc please see specific comments in the ms. This is to my opinion one of the most important caveats in the current version of the ms, but it should be easy to solve.

In the results section it will be useful to include a simple graphic with the densities of the different taxa. This kind of graphs are very informative and help the reader to compare in a glanz with results from other areas. Of course if % will be used it needs to be clearly included in the figure the N!

The discussion will highly benefit from a deeper literature research.

Comments to the suppl mat files:

- both tables should be part of the main text.

- Table S1: column Images annotated should be inlcuded beofre the column Number of images analysed

- Table S2: I will expect to see in this table the number of specimens counted in each image. This information is important!

I wish you good luck with the re-submission of the ms, but I am sure all the comments I did are easy to deal with!

Best,

Cova

Author Response

July 31st

Re: Submission of revised manuscript

Dear Reviewer 2,

Thank you very much for your positive evaluation of our manuscript entitled “Deep-sea epibenthic megafaunal assemblages of the Falkland Islands, South-west Atlantic" (diversity-1792620). We would also like to thank you for your very helpful feedback: we have addressed your suggestions in the revised manuscript.  A full list of our responses to your comments can be found in the attachment. In particular, as suggested we have expanded on the methodology for image collection and analysis in the methods. We have also included in the supplementary the number of each taxa annotated and density data in Table S2 and a stacked bar chart (Figure S3) to show the relative contribution of the different taxa to each assemblage. We thank you for your suggestion of including the morphospecies catalogue used in the study and we have included example images of characteristic morphospecies of each assemblage in the supplementary Figure S4. We thank you for the suggestion to consider introducing the South Atlantic prior to the Southwest Atlantic but feel that since the focus of this paper is the Southwest Atlantic, which in itself is a distinct region we would want to limit our introductory literature to the Southwest Atlantic. We address your comment regarding biased literature by including literature from the South East Atlantic, Mediterranean and Antarctic in the discussion.

We believe that this version of the manuscript now describes deep-sea epibenthic faunal assemblages of the Falkland Islands even better.

Sincerely,

Tabitha Pearman

Dr Tabitha R R Pearman

South Atlantic Environmental Research Institute

Stanley, Falkland Islands

Reviewer 3 Report

The submitted article by Pearman and collaborators is a valuable contribution to improve our knowledge on the biological diversity of the benthic ecosystem from deep-sea areas poorly explored in the south-west Atlantic. The authors take advantage of legacy images to provide the first quantitative analysis of the benthic assemblages found around the Falkland Islands, between 1000 and 1800 m depth. The article is well presented, with a very informative and well-written introduction and discussion, and besides some concerns listed below, which should not become too difficult to address, this manuscript is worthy of publication.

The introduction states the scientific knowledge gaps regarding deep-sea exploration in the southwest Atlantic and how this limits our capacity to manage its benthic ecosystem, and clearly sets the objectives of the study. The data collected is well treated and graphically well displayed, and in general well discussed. The methods section could be improved to better show how image collection was carried out, and could also provide further details regarding the data analyses to allow other researchers to replicate the workflow employed. The discussion is also well structured and referenced throughout, although I suggest introducing the patches with B. candida and the sponge aggregations as variants within Assemblage 1 rather than consider them separate assemblages based on the results of the RDA biplot (see further comments below).

One of the main suggestions that I would make to improve the utility of the work presented here relates to the identification of the benthic fauna. Since a large part of the taxa observed in the images have been classified in high taxonomic levels (Family, Order; Class, Phylum) and listed as morphospecies (for which only the authors have reference images), a photographic catalogue provided as supplementary material would be highly useful. This catalogue could provide a very important link between the morphospecies names given and the aspect of the organisms observed, which could serve as reference for comparison with other studies carried out in the region (or in the Atlantic in general). Furthermore, the limitations regarding taxa identification from benthic imagery in the area should be acknowledged in the discussion of the article, providing clues to what could/should be done in order to improve our capacity to identify the deep-sea fauna from imagery in an area that has been historically been poorly sampled.

 Specific comments to the different sections:

 Abstract:

-       - The year when the survey was performed should be indicated.

-       - The abstract indicates that images were collected between 1200 and 1800 m, while the results section provide a different depth range (1070-1880). Please check for consistency.

-       - The abstract indicates that 862 images were annotated but only the data referring to 286 ‘Sample’ images are provided in the results section. These values should be checked for consistency.

-       - The abstract should indicate the number of assemblages identified from the clustering analysis and also their characteristic species.

-       - Line 27-29. The authors state that the quantitative analyses provide a baseline to assess human impacts. Since no quantitative data is provided in the ms, I believe this is an overstatement, and authors should refer only to comparisons with other benthic assemblages.

Introduction:

-       - Line 42: add “is” to the sentence “our current knowledge of epibenthos is predominately based on ex situ data”

-       - Line 42: ex-situ

Materials and methods:

2.1 Study area

-       - Line 91: change “The Falklands Trough” for “the Falklands Trough”

-       - Line 105: the authors state that the Antarctic intermediate water (AAIW) is characteristic of depths below 1000 m, and then state that the Upper Circumpolar Deep Water (UCDW) is found between 1000-2200 m. This statement should be revised to avoid an overlap in the depths provided.

 2.2. Quantitative image analysis

-       - There is no reference to the surveys performed to collect the underwater images. Authors should mention the imaging platform used and its main technicalities (type of camera, images obtained as photographs or video stills, resolution of the images analyzed, use of underwater positioning system, presence of parallel lasers, method of survey) and how the images were extracted (stills from video or photographic images at set intervals).

-       - The dates when the surveys were conducted and the area (or distance) covered by the imaging platform over the seabed should also be provided.

-       - Line 121. There should be a reference to the annotation software employed (if any).

-       - Line 122. The way the area of each image was calculated is not explained. Were images taken vertically with respect to the seafloor and parallel lasers used for image scaling? Or were the images taken with an angle? If this was the case, how was the area of each image calculated? These aspects should be further clarified.

-       - Line 124: The authors indicate in line 119 that 862 images were annotated. Then, in line 124, the authors mention that ‘Sample’ images were annotated. Since images were probably not annotated twice, I suggest rewriting to something as: “‘Sample’ images were selected from the pool of annotated images at 60 second intervals”.

-       - How many “Sample” images were selected? What was the percentage of “Sample” images with respect to the total pool of annotated images? This value appears in the results section but it should be shown in the Materials and methods section.

-       - There is no reference as to why an interval of 60 seconds was selected. Would it not have been easier to extract images based on a set distance covered by the imaging platform? The distance between images collected at 60 seconds intervals will be highly dependent on the behavior of the platform during the dive (i.e. if stopped in the same spot for 5 minutes, 5 ‘Sample’ images will be of the same exact location), as well as differences in the average speed of the platform in different dives. These aspects should be further clarified.

-       - Line 127. The authors should provide a reference to where the EUNIS 2022 classification system can be found. Further, a list of the substrate categories employed in this study could be provided, together with a short description. A figure displaying the main types, maybe provided as a supplementary figure, would also be helpful.

-       - Regarding substrate composition per image. How was substrate estimated? As a percentage of the different EUNIS categories based on a superimposed grid on each image? Or was only the dominant substrate considered? These aspects should be further described in the text.

-       - Lines 128-133. The authors provide a list of oceanographic variables that were not used in the statistical analyses. I suggest that Table 1 provides only the list of environmental variables that were actually incorporated in the multivariate analyses, even if the authors explain in the main text that several other oceanographic variables were available (providing examples) but not used due to a coarse resolution or a high collinearity with other variables.

2.3 Data analysis

-       - Line 142. The authors state that images with less than 3 taxa were removed from the analyses. If this statement refers to images with less than 3 organisms, I suggest replacing “taxa” for “organisms”. If the authors refer to images where less than 3 morphospecies were observed regardless of their abundance, how did they deal with assemblages that displayed a dominance of just 1 or 2 species with high densities? Were these images also removed from the analyses?

-       - Line 145-167. The set of multivariate analyses performed could be described in better detail. I suggest associating to each of the analyses the function and R package used (or PRIMER routine), allowing the reader to replicate the workflow employed by the authors.

Results:

-       - Lines 169-172: the following sentence should be moved to the Materials and Methods section: “Benthic imagery was obtained from 69 stations collected during three environmental baseline surveys in 2014 (See supplementary) (Falkland Islands Government Department of Mineral Resources, unpublished data) that covered water depths of 1070 m to 1880 m.”

-       - Line 172: If 862 images were annotated, why do the results provided refer only to the data collected in the 286 ‘Sample’ images? Could the general results have a broader perspective and include all the data gathered from the full annotation, and then show the community analysis based on the annotation of the ‘Sample’ images? It feels like the large effort placed in annotating all the images is not used to show the diversity and abundance of species that are present in the area.

-       - According to Supplementary Table S2, only 16 taxa were identified to genus/species level, and the remaining (morpho)species were left in higher taxonomic levels. This limitation in the taxonomic resolution of the fauna annotated should be reflected in the text of the Results section.

-       - Line 174-175: “Many were rare, and only 17 morphospecies were observed from more than 10% of sample images (See supplementary)”. The authors refer to Figure S1 here. I believe these results should not be graphed, at least in the way they are graphed in Fig. S1 (see below). I think the best way to illustrate the rarity of most species would be to add an extra column in Table S2 to show the percentage of images in which each of the morphospecies was observed. Also, it would be very useful to add one extra column with the total number of observations for each taxa in all the images annotated and also the number of observations per assemblage, replacing the signs used to determine presence/absence. This value would provide a better idea of what species/taxa are more prevalent and abundant in the images collected in the study area.

-       - Lines 175-177. The authors could refer in the actual text to other abundant and/or frequently observed species, instead of just providing data for the most abundant species of all. This would provide a better picture of the benthic diversity observed in the area, and reflect the effort placed in the annotation of the images.

 3.1. Quantitative analysis of epibenthic megafaunal assemblages

-       - Line 192: “Five clusters were identified from the hierarchal clustering and nMDS plot”. As I understand, the number of clusters into which divide the dataset (five in this case) was determined by the results of the silhouette analysis, and ‘Sample’ images were assigned to each group by means of a hierarchical clustering. Then, the assignation of each sample to a group was superimposed over the results of the nMDS. Line 192 should be modified to better explain how groups were identified and ‘Sample’ images assigned.

-       - The Figure with the hierarchical clustering could be provided in the Supplementary material to show the affinities between groups.

-       - Authors mention in the materials and methods that species counts were converted into densities per m2. There is no reference in the Results section neither in the Supplementary to species densities. If these values were calculated, authors could provide estimates of species densities within the assemblages identified.

-       - Line 196: remove the “s” in “represents”

-       - The description of each assemblage could be expanded, with information on the abundance (or density) of the main species and also their depth range and main substrate types.

3.2. Environmental drivers of faunal assemblages

-       - As I understand from the RDA, substrate type correlates very well with current velocity along the first axis, and finer sediments are found in areas of less current activity and harder substrates in areas of higher current intensities. I see depth as the main factor along the second axis, not aspect or current velocity. 

-       - The authors provide a map with the distribution of the assemblages in the study area (Figure 4). This map could be briefly described in the Results section.

Discussion:

4.1. Deep-sea epibenthic megafaunal assemblages of the Falkland Islands

-       - Lines 267-269: The authors indicate that the analysis performed identified four main assemblages characterized by fragile habitat-forming taxa that can be considered indicators of VMEs. Since the silhouette and cluster analyses determined 5 groups, there should be a reference to why the remaining assemblage does not fall within the definition of what constitutes a VME.

-       - Lines 270-271: The authors indicate that the RDA analysis performed allows for an extra assemblage to be identified based on the results of the biplot. Since this extra assemblage was not determined by the silhouette analysis and hierarchical clustering, I suggest that authors indicate that, although B. candida was not identified as a characteristic species of Assemblage 1, it was observed forming reef-like structures that are of ecological importance in the area. And from here, authors can develop the discussion about the importance of finding B. candida reefs in the study area, including the capacity to provide surface for attachment to other species as well as refuge areas for smaller taxa to hide (which I think is well explained and documented in lines 272-295).

-       - The following sentence should be revised: “The B. candida assemblage that was differentiated from within cluster one by the RDA analysis (Figure 5), was observed in association with reef like aggregations of B. candida.”

-       - Line 273: remove “n” from “South American”

-       - Lines 305-308: Same as above. I suggest not considering the species listed in those lines as a separate assemblage from Assemblage 1 based on the results of the RDA since it was not depicted as a different group from the hierarchical clustering. Rather, the authors could explain that there were areas where some of the characteristic species of Assemblage 1 showed a higher dominance and higher abundances. And then develop the rest of the discussion on sponge fields from here.

-       - Line 339: Change “Hormathia pectinata” for “Hormathia pectinate

Figures:

Figure 1:

-       - The width of the lines delimiting the extent of FICZ and FOCZ could be increased for an easier interpretation of the limits of the conservation areas

-       - The main isolines (the ones around the Falkland Islands) could have the depth superimposed. This would provide an easier interpretation of the areas in which the underwater surveys were carried out

-       - The map would be clearer if 3 insets were added (zoom ins of the three main areas where the images were collected).

-       - Since the latitudes shown are 50º, 60º and 70º, minutes (0,000’) could be removed.

Figure 2:

-       - Since the main objective of this study is to identify species assemblages, and to make the interpretation of the results easier for the reader, I suggest that this figure is redone, showing one characteristic image from each of the assemblages identified in the clustering analysis labelled from 1 (A) to 5 (E) in the order provided in Table 2. And if there are variations of one particular assemblage (as explained in the discussion), this can also be referenced, always maintaining the order provided in Table 2. According to how the figure is structured now, image (A) corresponds to Assemblage 3, image (B) has an aggregation of a species that does not relate to any of the characteristic species listed in Table 2, but according to the RDA is part of Assemblage 1, image (C) has species from Assemblage 2 and 3, image (D) corresponds to Assemblage 1, image (E) also has species from Assemblage 2 and 3, and image (F) corresponds once again to Assemblage 1. There are no images of Assemblage 4 and 5, and some images have species that correspond to more than one assemblage.

-       - Specific images could be referenced in the text next to the descriptions of each assemblage (Section 3.1) as “Figure 2A-B”, “Figure 2C”, “Figure 2D”, etc. This would facilitate the identification of each image with respect to the assemblage being described in the text.

-       - Images could incorporate a scale bar as a reference for the size of the organisms.

-       - Arrows in the images could be useful to indicate the taxa/morphospecies being referred to in the figure caption.

Figure 3:

-       - The main species that determine the distribution of the ‘Sample’ images in the biplot could be superimposed, in order to show what species are characteristic in each assemblage.

-       - Hellinger is the distance measure used to compute the dissimilarity matrix, so the caption should be written something like “non-metrical Multidimensional Scaling (nMDS) over a Hellinger distance matrix of transformed species abundance data”

-       - In any case, the type of transformation applied to the species abundance or density data is not provided in the Materials and Methods.

Figure 4:

-       - The caption should provide an indication to the areas delimited by the green and red lines.

Figure 5.

-       - Captions for plots (A) and (B) are mixed up.

-       - The caption states that species data is transformed but this information does not appear in the methods. What type of transformation was applied?

-       - Hellinger is the distance used. I suggest: “Plots of the Redundancy Analysis (RDA) computed over a Hellinger distance matrix from transformed species abundance data and selected environmental variables”

-       - The sentence “The vector arrowheads represent high, the origin averages, and the tail (when extended through the origin) low values of the selected environmental variables.” could be removed since vectors in canonical analyses work in the same way.

-       - The sentence “Sites close to one another tend to have similar faunal composition that those further apart.).” could also be removed since it does not add any significant information to the figure caption.

Tables:

Table 1:

-       - I suggest providing a list of the actual variables used in the statistical analyses performed in this study. Since the remaining variables were not used at all, there is no need to show them in this table.

Table 2:

-       - The substrate categories listed here have not been explained in the main text, and there should be a reference to what the categories stand for, especially “Mixed”, “Coarse” and “Hard”.

-       - Flabellum should be in italics

-       - Maybe the cut-off level of the SIMPER analysis could be modified in order to include more species as characteristic for each group? The species B. candida, T. viridis, and M. spinosa are common in Assemblage 1 but do not appear in this table.

Supplementary

Table S1

- Specify that positions are given in decimal degrees (º)

- The date when each of the dives was performed and the underwater imaging platform used should appear on the table

- There should be an explanation of what is the difference between “images analysed” and “Images annotated” on the table caption. This difference is not well explained.

- Why are the station locations “proposed”?

Figure S1

- I find Figure S1 difficult to understand. As I gather from the results section, the graph shows that 140 species appear in 0-10% of the samples, 17 species in 10-20%, 5 in 20-30% and so forth. I do not think these type data should be graphed in the way it is done, and I think it would be sufficient to explain these results in the main text. Frequency distributions are generally displayed with the percentage of observations on the y axis, and what is being represented along the x axis.

 Table S2

- Morphospecies could be grouped by phylum instead of having all taxa ordered alphabetically.

- Morphopecies should be ordered according to the morphospecies number provided (e.g. “Alcyonacea.sp.10” and “Alcyonacea.sp.19” should appear after “Alcyonacea.sp.2”)

- Points could be removed and a space added before “sp.”: e.g. “Actiniaria sp. 2” instead of “Actiniaria.sp.2

- The depth range of each morphospecies as observed in the images could also be provided.

- As commented above, this Table could contain the total number of organisms annotated for each morphospecies, as well as within each cluster. This would provide an indication of their abundance in the area and also within the assemblage, instead of just showing their presence in the assemblage as indicated by a sign.

- The caption should refer to the number of still images that were annotated to produce the results of Table S2.

- The table includes 157 taxa, but only 16 have been identified to genus or species level (10%). As commented above, since the article is of a descriptive nature, it would be extremely useful that the species listed here are accompanied by a photographic catalogue including an image of each of the taxa listed. This would facilitate comparisons with other areas based on data collected from past (and future) surveys.

Author Response

July 31st

Re: Submission of revised manuscript

Dear Reviewer 3,

Thank you very much for your positive evaluation of our manuscript entitled “Deep-sea epibenthic megafaunal assemblages of the Falkland Islands, South-west Atlantic" (diversity-1792620). We would also like to thank you for your very helpful feedback: we have addressed your suggestions in the revised manuscript, which you can find in attachment.  A full list of our responses to the  comments can be found in the corresponding comments that are attached. In particular, as suggested we have expanded on the methodology for image collection and analysis in the methods. We have also included in the supplementary the density data in Table S2 and a stacked bar chart (Figure S3) to show the relative contribution of the different taxa to each assemblage. We thank you for your suggestion of including the morphospecies catalogue used in the study and we have included example images of characteristic morphospecies of each assemblage in the supplementary Figure S4. We have removed supplementary Figure S1 and incorporated information regarding the occurrence of taxa across samples in the revised supplementary Table S2. We have also added additional figures to the supplementary to provide examples of annotated substrata categories (Figure S1), the results from the hierarchal clustering (Figure S2) and we have amended Figure 2 as suggested by changing the order of images to match the order of clusters described in Table 2.

We believe that this version of the manuscript now describes deep-sea epibenthic faunal assemblages of the Falkland Islands even better.

Sincerely,

Tabitha Pearman  

Dr Tabitha R R Pearman

South Atlantic Environmental Research Institute

Stanley, Falkland Islands
